# Benchmarking Zero-Shot Recognition with Vision-Language Models: Challenges on Granularity and Specificity

## Abstract

This paper introduces innovative benchmarks to evaluate Vision-Language Models (VLMs) in real-world zero-shot recognition tasks, focusing on the pivotal properties of granularity and specificity. We propose a unique evaluation protocol using adapted ImageNet and MS-COCO datasets to assess models' consistency in recognizing concepts at varying granularity levels and their sensitivity to the specificity of language inputs. Our extensive evaluation reveals that state-of-the-art VLMs, including contrastive models like CLIP, struggle with granularity and are sensitive to text specificity, impacting their effectiveness in open-world settings. This comprehensive study, a first in evaluating VLMs from these perspectives, provides valuable insights and tools for the community, highlighting the limitations and paving the way for enhanced models with better generalization in zero-shot recognition. Our benchmark will be open-sourced upon acceptance.

## 1 Introduction

Vision-language models (VLMs) have shown impressive capabilities in a wide range of tasks, including image captioning (Wang et al., 2022; Yu et al., 2022), visual question answering (Chen et al., 2023), and notably, zero-shot visual recognition (Radford et al., 2021; Zhou et al., 2022; Gu et al., 2021; Xu et al., 2022). Models pretrained on large-scale image-caption datasets (Jia et al., 2021; Yuan et al., 2021; Schuhmann et al., 2021; 2022a), like CLIP (Radford et al., 2021), have been at the forefront. They map visual and language inputs to a shared latent space and the cross-modality score indicates the how well the visual input matches the language defining any concept. This zero-shot capability, enabling the recognition of unseen objects or concepts without additional training, is crucial and desired towards building general intelligent visual systems.

An ideal open-world zero-shot model, should recognize any language-defined input, from simple concepts like "an image of flowers" to more complex descriptions like "a person playing with a dog on the beach", and output scores indicating whether the visual input *correctly* imply the semantics of the language input. Existing works often evaluate the zero-shot capability on classification dataset like ImageNet (Russakovsky et al., 2015) and a collection of domain specific classification datasets (Li et al., 2022a) without the notion of granularity of concepts, as well as image and text retrieval on Flickr30K (Plummer et al., 2015) and COCO (Lin et al., 2014) that are not able to reveal the general failure pattern. These benchmark fall short of replicating the complexities of a realistic open-world setting, leaving a substantial gap in our understanding of the effectiveness of VLMs in such scenarios.

This paper present new benchmarks on the pivotal properties when deploying VLMs for real-world zero-shot recognition: *granularity* and *specificity*. First, VLMs should have consistent understanding of concepts at different *granularity* level. Take an example shown in Figure 1-Left, if a model can successfully recognize an image of a leopard given the language query of a fine-grained concept, such as "an image of a leopard", it should also recognize it with a query of coarse-grained concepts, like "an image of a feline". The consistency across semantic granularity not only indicates whether a model truly grasps the relationships between concepts but also is crucial for applications. For instance, in the context of autonomous driving perception, it's concerning if a model can recognize an image of a road cone using the query "road cone" but fails with "barrier". Second, we evaluate how the specificity of language inputs can effect the outputs of VLM despite if the language and visual

Figure 1: **Left**: Zero-shot models should recognize images with fine-grained (FG) concepts such as "Leopard", as well as coarse-grained (CG) concepts like "Feline" However, they often exhibit performance discrepancies on concepts at different levels of granularity. **Right**: Zero-shot models should recognize whether the text correctly describe the given image. However, vision-language models could be sensitive to the specificity of text and struggle to distinguish between the challenging positive like single-label prompts and hard negatives like poisoned captions with small changes.

inputs matches. As Figure 1-right shows, a simple prompt "a picture with a dog" get a lower score than a more specific caption "a dog and cow lying together on an orange couch" while the former one is a correct description and the latter one is wrong with a lot correct details. This benchmark fundamentally reveals whether VLMs can faithfully reflect the correctness between the visual and language inputs rather than only holistically similarity.

To fairly test the *granularity* consistency, we propose an evaluation protocol on the task of recognizing a coarse-grained class by measuring the performance discrepancy between directly use the prompt of the coarse-grained class and aggregating the predictions from prompts of its fine-grained children classes. A dataset with hierarchical labels is essential for this evaluation and therefore we adapt ImageNet along with its semantic hierarchy from WordNet. To test the *specificity* robustness, we use image-to-text retrieval task on the MS-COCO Lin et al. (2014) by designing hard positive text with different specificity e.g. single-label prompts with less information, as well as hard negative text, e.g. wrong captions with a small modification.

With the carefully designed benchmark, we extensively evaluate state-of-the-art VLMs, particularly contrastive models like CLIP, that covers various factors including pretraining datasets, architecture, cross-modality designs, and learning objectives, and find that vision-language models struggles at both benchmark.

In the granularity evaluation, we find that *VLMs are rather better at recognizing moderately fine-grained concepts than high-level coarse-grained concepts and the training data may account for the observed behavior*. We further analyze the distribution of concepts at different granularities in LAION dataset and find the moderately fine-grained concepts are more presented in image alt-text. The positive correlation between the frequency gap and performance discrepancy demonstrate the impact of data distribution. In the specificity evaluation, we found that *VLMs are sensitive to the specificity of text*: correct text whose specificity are different the training data, e.g. simple single label prompts or overly long captions, may produce lower scores than the captions with correct specific details but small errors. As a consequence, retrieving hard positive text from hard negative text is extremely challenging, which suggests that the scores of VLMs do not indicate correctness of the text regarding to visual inputs faithfully. Fine-tuning with hard text can improve the performance on the benchmark but may not be a complete solution.

To our best knowledge, this is the first comprehensive study that evaluating VLMs from the perspective of semantic granularity and specificity. We believe that the carefully designed benchmark provide a valuable tool to the community to better quantitatively evaluate VLMs. With the proposed benchmark, we observed that all models surprisingly performs significantly worse than what we may hope. The findings and insights from our analysis may shed lights on better understanding the limitations of current VLMs and the challenges of using it for zero-shot recognition, and inspire new models with better generalization.

## 2 RELATED WORKS

**Zero-shot visual recognition**    CLIP-like vision-language foundation models have enabled open vocabulary visual recognition by mapping images with their corresponding language descriptions. Early methods (Radford et al., 2021; Jia et al., 2021) demonstrate the effectiveness of this paradigm

on the image classification tasks. For example, CLIP is able to achieve decent zero-shot accuracy on 27 image classification datasets. Given its potential, the language-driven visual recognition paradigm has been extended to tasks including object detection (Zhou et al., 2022), semantic segmentation (Xu et al., 2022), video action recognition (Wang et al., 2021), depth estimation (Zhang et al., 2022), etc. Such language guided visual recognition has become the new paradigm in the field of computer vision since it can recognition new objects without any training data. In this paper, we would like to stress test these VLMs in terms of zero-shot visual recognition to better understand their capability and limitation in realistic open-world setting.

**Benchmarking vision-language models** Thanks to the larger datasets and larger transformer models, many powerful vision-language models have been developed and shown great capability (Yu et al., 2022; Wang et al., 2022; Chen et al., 2023). At the same time, these models are being studied from various perspectives, such as robustness, bias, and other limitations (Galindo & Faria, 2021; Fort, 2021; Goh et al., 2021; Noever & Noever, 2021; Daras & Dimakis, 2022). Qiu et al. (2022) investigates the robustness of nine open-sourced image-text models under common perturbations on five tasks, while Schiappa et al. (2022) studies the robustness of video-text models. Fang et al. (2022) further analyzes the robustness of VLMs under challenging natural distribution shifts and show that the more diverse training distribution is the main cause for the robustness gains. Yuksekgonul et al. (2022); Thrush et al. (2022) systematically evaluates the ability to encode compositional information of the VLMs. Cho et al. (2022) investigates the visual reasoning capabilities and social biases of different text-to-image models. To improve transferability, Shen et al. (2022) designs an efficient and scalable approach that leverages external knowledge to learn image representations. In this paper, we study VLMs from two new perspectives: granularity and specificity through the lens of zero-shot visual recognition.

Table 1: An overview of the differences between the vision-language models evaluated in our study by the architecture, pretraining datasets, learning objectives, and if using cross-modality fusion modules. ITC, ITM, MIM, MTM, MMM stands for image-text contrastive, image-text matchng, masked image modeling, masked text modeling and masked multimodal modeling losses.

| Model | Architecture | Datasets | Objectives | Fusion |
|---|---|---|---|---|
| CLIP-B | ViT-B-32 | Private400M | ITC | - |
| OpenCLIP$_{B-400M}$ | ViT-B-32 | LAION400M | ITC | - |
| OpenCLIP$_{B-2B}$ | ViT-B-32 | LAION2B | | |
| OpenCLIP$_{L-2B}$ | ViT-L-14 | LAION2B | | |
| OpenCLIP$_{H-2B}$ | VIT-H-14 | LAION2B | | |
| UniCL$_{YFCC}$ | Swin-B | YFCC14M | ITC | - |
| UniCL$_{IN21K}$ | Swin-B | IN21K | | |
| UniCL$_{YFCC+IN21K}$ | Swin-B | IN21K+YFCC14M | | |
| UniCL$_{All}$ | Swin-B | IN21K+YFCC14M+GCC15M | | |
| K-LITE | Swin-B | IN21K+YFCC14M+GCC15M | ITC | - |
| BLIP | ViT-B-16 | COCO+VG+CC+SBU +LAION+CapFilt-L (129M) | ITC + ITM + Captioning | - |
| BLIP$_{ft-coco}$ | | | | - |
| BLIP$_{ft-coco \& fusion}$ | | | | ✓ |
| FLAVA | ViT-B/16 | PMD70M | ITC+ITM+MMM+MIM+MTM | - |

# 3 ZERO-SHOT VISUAL RECOGNITION WITH VISION AND LANGUAGE MODELS

In this study, we focus on two-stream contrastive vision-language models, such as CLIP, which leverage contrastive pre-training on a large dataset of paired image-text samples to learn cross-modal alignment. These models typically consist of a visual encoder $E_v$ and a text encoder $E_t$, for encoding visual inputs $x_v$ and textual inputs $x_t$ into aligned representation spaces.

The zero-shot visual recognition task with a vision-language model can be formulated as computing the cross-modality score:

$$f(x_v, x_t) = E_v(x_v) \odot E_t(x_t) \tag{1}$$

Here, the $\odot$ operator computes the score between visual and language embeddings, with cosine similarity being the common choice while some models like FLAVA use an additional module to

fuse the multi-modal embeddings. For classification tasks, $x_t$ can be a class prompt, such as 'a photo of a car," or it can incorporate additional class-specific knowledge to improve performance. In our subsequent studies, we adopt the prompt templates used in Radford et al. (2021) for classification tasks. We simplify $E_t(x_t)$ and $f(x_v, x_t)$ for a class $y$ to $E_t(y)$ and $f_{cls}(x_v, y)$, respectively.

In our study, we evaluate various contrastive vision-language models, each with distinct backbone architectures, training data, and learning objectives, shown in Table 1. These variants include CLIP (Radford et al., 2021), OpenCLIP (Ilharco et al., 2021) (trained on the public LAION dataset Schuhmann et al. (2022b)), UniCL (Yang et al., 2022) (which incorporates classification annotations into the contrastive learning objective), KLITE (Shen et al., 2022) (which augments alt-text with extra knowledge during training), FLAVA (Singh et al., 2022) (trained with both cross-modal and uni-modal data and losses), and BLIP (Li et al., 2022b) (which includes uni-modal and cross-modal training, along with a captioning head for data bootstrapping). By examining these models, we aim to gain insights into the zero-shot visual recognition capabilities of vision-language models.

# 4 EVALUATE GRANULARITY CONSISTENCY

In this section, we study whether vision-language models (VLMs) performs consistently on visual concepts at different at different levels of granularity, which also indicate if the model inherently understand the relationship between concepts. We build a benchmark to quantitatively evaluate VLMs on the performance discrepancy between concepts at different granularity level. Intuitively, for a vision-language model that understand that "feline" includes "lion", "tiger", "leopard" and so on, its performance of directly recognizing if a image is "feline"should be consistent with the performance by aggregating the results from recognizing if an image is "lion", "tiger", "leopard" and so on. Our results shows that all models trained on image-caption pairs shows significant performance discrepancy and the models recognize better with moderately fine-grained concepts. The further analysis shows that the distribution of training data may account for the discrepancy.

## 4.1 MEASURE PERFORMANCE DISCREPANCY ON A SEMANTIC HIERARCHY

To evaluate the understanding of vision-language models across concepts at different semantic granularities, we employ zero-shot classification as the task. However, directly comparing classification metrics between classes at different granularities is not appropriate since recognizing fine-grained classes is inherently more challenging. Instead, we measure the performance discrepancy in zero-shot classification between directly predicting with prompts of coarse-grained (CG) class names verse propagating predictions using prompts of their finer-grained (FG) children classes.

**Dataset** We build on a dataset with multi-level label hierarchy by expanding ImageNet-1K dataset. We assign each of the 1000 leaf fine-grained label with its ancestor labels based on the label hierarchy derived from WordNet, resulting in additional 820 ancestor labels. For example, "leopard" images are also labeled as "big cat," "feline," "mammal," "animal," and so on. This dataset allows us to evaluate the performance discrepancy of concepts at different granularities.

**Evaluation Protocol** In a dataset with label hierarchies, each image has multiple labels and therefore the proxy task is a multi-label classification task that each label is classified independently as a binary classification. For coarse-grained ancestor labels, the classification can be done by directly using its own class name or propagating predictions on its fine-grained children labels. Propagating predict labels requires choosing score thresholds calibrated on labeled data. We use averge precision (AP) as the metric which only requires raw predicted scores and avoid selecting threshold. We report the performance discrepancy for each ancestor label individually by measuring the performance difference between using raw scores and propagated scores from its children labels. For example, given an input image, its score of label "feline" can be got alternatively by propagating its scores of label "lion", "tiger", "leopard" and so on. We formulate the protocol below.

For a class $y$, the raw cross-modality score between an image $x$ and the textual prompt of $y$ is computed by $S^{raw}(y) = f(x, y)$. For an ancestor class $y_i$, we designed two ways to aggregate scores from its direct children classes $Y_C^i$. The two ways are visually illustrated in Figure 2 and their corresponding formulations are provided below.

Table 2: Zero-shot multi-label classification performance of labels at different levels of granularity on ImageNet. We reported the mean average precision (mAP) of ImageNet-1K fine-grained classes (leaves), and their coarse-grained ancestor classes with raw predictions (Ancestor$_\text{raw}$) and two propagated predictions. The differences ($\Delta$) between the raw and propagated performance of ancestor classes presents the performance discrepancy of vision-language models on concepts at different granularity. Propagating from leaf classes gives the best performance.

| Model | Leaves | Ancestor$_\text{raw}$ | Ancestor$_\text{child}$ ($\Delta$) | Ancestor$_\text{leaf}$ ($\Delta$) |
|---|---|---|---|---|
| CLIP-B | 50.10 | 24.91 | 45.35 (+20.44) | 58.73 (+33.83) |
| CLIP-L | 65.06 | 33.64 | 57.72 (+24.08) | 72.25 (+38.61) |
| OpenCLIP$_\text{B-400M}$ | 47.10 | 20.12 | 40.66 (+20.54) | 54.50 (+34.38) |
| OpenCLIP$_\text{B-2B}$ | 54.97 | 24.95 | 47.64 (+22.69) | 62.66 (+37.70) |
| OpenCLIP$_\text{L-2B}$ | 65.79 | 31.59 | 56.65 (+25.07) | 72.53 (+40.94) |
| OpenCLIP$_\text{H-2B}$ | 68.28 | 32.70 | 58.70 (+26.00) | 74.93 (+42.23) |
| UniCL$_\text{YFCC}$ | 35.75 | 20.13 | 35.90 (+15.77) | 47.55 (+27.42) |
| UniCL$_\text{IN21K}$ | 26.28 | 38.15 | 39.30 (+1.15) | 41.23 (+3.08) |
| UniCL$_\text{YFCC+IN21K}$ | 37.84 | 35.18 | 44.84 (+9.65) | 51.55 (+16.37) |
| UniCL$_\text{All}$ | 54.49 | 37.54 | 54.58 (+17.04) | 65.85 (+28.32) |
| K-LITE | 48.40 | 31.50 | 49.63 (+18.14) | 61.58 (+30.08) |
| BLIP | 41.87 | 20.31 | 39.44 (+19.13) | 52.08 (+31.77) |
| BLIP$_\text{ft-coco}$ | 42.83 | 22.07 | 41.45 (+19.38) | 54.00 (+31.93) |
| FLAVA | 40.91 | 21.36 | 39.32 (+17.96) | 51.89 (+30.53) |

1. Propagate from direct children: maximum of the *raw* scores of the direct children classes.

$$S^\text{child}(y_i) = \max_{y_j \in Y_C^i} S^\text{raw}(y_j) \tag{2}$$

2. Propagate from leaves: maximum of the *aggregated* scores of the direct children classes, which is equivalent to the maximum of the raw scores of its leaf children classes.

$$S^\text{leaf}(y_i) = \max_{y_j \in Y_C^i} S^\text{leaf}(y_j) \tag{3}$$

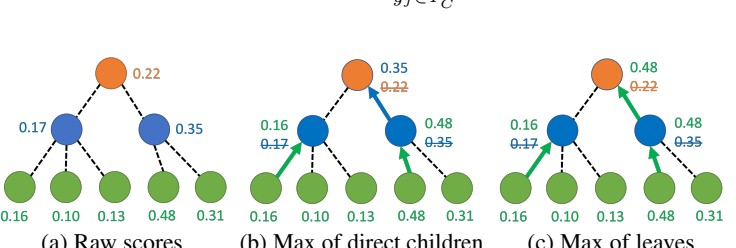

(a) Raw scores     (b) Max of direct children     (c) Max of leaves

Figure 2: Illustrations on the two ways to propagate scores on the semantic hierarchy. (a) Raw scores without propagation. (b) Propagate the max score from direct children classes. For example, 0.35 = max(0.17, 0.35) (c) Propagate the max score from leaf classes. For example, 0.48 = max(0.16, 0.10, 0.13, 0.48, 0.31)

## 4.2 RESULTS AND ANALYSIS

We tested different vision-language models on the granularity benchmark and the results are shown and Table 2. Note that the $\Delta$ columns in the table presents the performance discrepancy of vision-language models on concepts at different granularity. We summarize our observations below.

**Vision-language models perform better on moderately fine-grained concepts** Across all the tested vision-language models, we consistently observe that the direct predictions with coarse-grained (CG) labels is significantly worse than predictions propagated from fine-grained (FG) labels. For

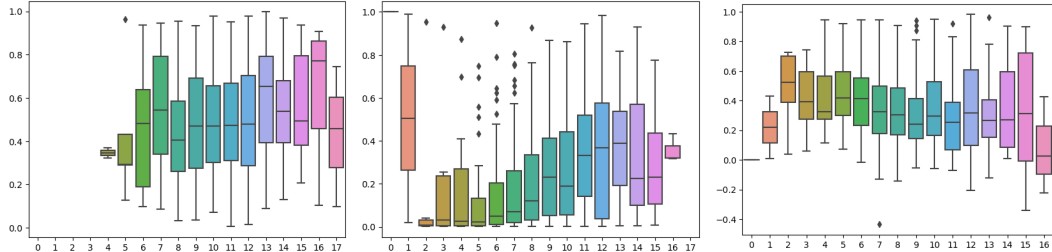

Figure 3: **Left**: The box-plot of zero-shot classification performance (mAP) for leaf class over the level in the semantic hierarchy. **Middle**: The box-plot of classification performance (mAP) for ancestor classes over the level in the semantic tree. Note that level 0 and level 1 have 1 and 2 classes respectively and easy to get high mAP. **Right**:The box-plot of *improved* zero-shot classification performance (mAP) for ancestor class by propagating from leaf classes, over the level in the semantic tree.

different propagation strategy, propagating scores from the leaf (the most fine-grained) classes works the best and significantly outperform propagating from the direct children. These results shows that vision-language models generate more reliable output when prompted with finer-grained concepts.

**Impact of Pre-training Data:** By comparing models trained on different data sources, we find that the distribution of training data, rather than its scale, plays a significant role in the performance discrepancy between CG and FG concepts. UniCL models were training on both classification dataset, ImageNet-21K (IN21K), as well as alt-text data YFCC-14M and GCC-15M.

- Comparing the UniCL models trained on IN21K and YFCC14M, we see that $UniCL_{YFCC}$ trained on image alt-text data achieves significantly better direct classification performance on FG leaf labels than $UniCL_{IN21K}$ (35.75 vs. 26.28 mAP). $UniCL_{IN21K}$ performs better on CG ancestor$_{raw}$ performance (38.15) due to the inclusion of all CG classes in IN21K, surpassing CLIP and OpenCLIP trained on alt-text data with a much larger scale. As a result, $UniCL_{IN21K}$ exhibits the smallest CG-FG discrepancy.

- Adding YFCC14M and GCC15M to IN21K for training UniCL models leads to significant improvement in raw FG classification while causing a slight degradation in raw CG classification, resulting in a larger CG-FG performance discrepancy.

- Larger models trained on LAION2B, e.g. $OpenCLIP_{H-2B}$, have worse CG ancestor$_{raw}$ performance and larger discrepancy than $UniCL_{IN21K}$. It demonstrate that simply scaling up the alt-text training data or model sizes is not an effective solution to resolve the discrepancy.

**Performance at Different Granularity Levels:** We analyze the distribution of raw performance of leaf classes, ancestor classes, as well as the performance discrepancy, based on the levels in the label hierarchy, as Figure 3. As Figure 3-Left shows, leaf classes at higher levels tend to exhibit better performance, indicating that higher-level leaf classes are more reliably recognized by the vision-language models. However, we observe a significant drop in performance for leaf classes at the deepest level (level 17). We believe that the classes at level 17 include more extreme fine-grained concepts that can be rare in the training data. Also note that the more fine-grained classes are naturally more challenging due to the their portion in the benchmark. Therefore, propagation from leaf classes consistently improve the performance of most (775 out of 820) CG ancestor classes except for those at level 16 that may not benefit from the low-performed children classes at level 17, as shown in Figure 3-Right.

**Granularity bias in pretraining data** From on the above observations, we believe that the distribution of concepts in alt-text data, biased towards fine-grained concepts, contributes to the observed performance discrepancy. The habit of using precise concepts in natural language descriptions might be a driving factor behind this bias. Therefore, on the OpenCLIP models and its training data LAION-2B, we further study the distribution of visual concepts in alt-text data and its connection with the granularity discrepancy. We first use the ImageNet samples of each leaf class to retrieve

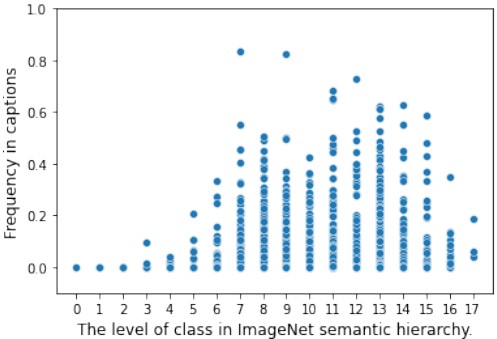 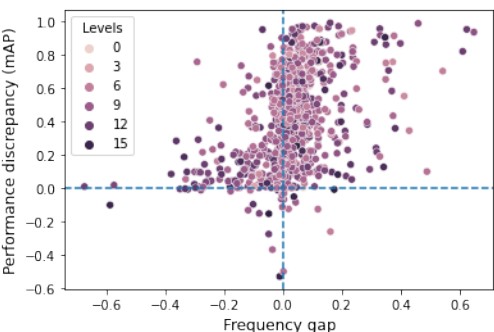

Figure 4: **Left**: The scatter-plot of the frequency of class names in pre-training captions over the level in the semantic tree. Course-grained and overly fine-grained concepts are less presented in captions. **Right**: The scatter-plot of performance discrepancy over the frequency gap between ancestor class names and their leaf children. A positive correlation exists between the performance discrepancy and frequency gap (coefficient 0.43 with p-value 3.4e-39).

similar images in LAION-2B and get $n_i$ images for $i$th leaf class. The frequency of $i$th class $q_i$ is defined as the ratio between $m_k$: the number of captions containing the name of $i$th class, and $n_i$. For $j$-th ancestor class, $m_j$ and $n_j$ are the summation of the $m$s and $n$s of their leaf children respectively. The frequency of class names in retrieved captions are plotted over their levels in the semantic tree as Figure 4-Left showed. The fine-grained classes (with higher levels) tend to have higher frequency in training captions although the overly fine-grained classes (level$\geq$16) are less presented in captions, matching the results in Figure 3 . We also study the correlation between performance discrepancy $\Delta_{\text{leaf}}$ of each ancestor class and its frequency gap with its leaf children $\Delta_{\text{freq}}$. For $j$-th ancestor class with leaf children $C_j$, the frequency gap is measured by $\Delta_{\text{freq}}^j = (\sum_{i \in C_j} n_i - n_j)/m_j$ and a positive gap indicate the preference over fine-grained class names. The overall distribution between $\Delta_{\text{leaf}}$ and $\Delta_{\text{freq}}$ is shown in Figure 4-Right. The positive ranking correlation (coefficient 0.43 with significant p-value $3.4e - 39$) also indicate the distribution of caption data accounts for granularity bias.

## 5 EVALUATE SPECIFICITY ROBUSTNESS

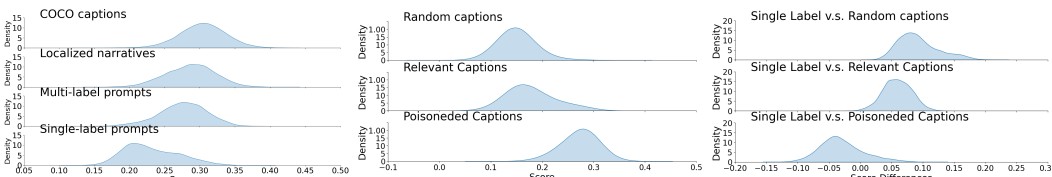

Figure 5: **Left**: The distribution of cross-modality score with positive text:COCO captions, Localized-narratives captions, single-label prompts, multi-label prompts. Unmatched specificity of text (lower or higher than normal captions) lead to low scores. **Middle**: The distribution of cross-modality score with negative text: captions from other random images, captions from relevant images and captions poisoned with small modifications. The poisoned captions get high scores comparable to positive text. **Right**: The distribution of the score differences between single-label prompts and various negative text. The correct single-label prompts often produce lower score than wrong poisoned captions.

When using vision-language models for open-world zero-shot recognition, the textual describes the visual concepts to recognize and the output score should indicate the chance that the described concepts exist in the visual input. In other words, it is critical to measure the correctness of textual inputs given visual inputs. However, as the example in Figure 1-Right illustrates, the scores of visual language models and do not strictly reflect the correctness of the textual input and thus make it challenging to be useful for open-world visual recognition. Since contrastive vision-language models have been trained on image alt-text pairs, the scores are biased toward the specificity of text as in the

Table 3: The performance of image-to-text retrieval task measured by mean Average Precision (mAP). Different columns represent different choices of positive and negative texts. $Cap^+$, $Cap_{ln}^+$, $Prompt_s^+$ and $Prompt_m^+$ stands for true COCO captions, Localized-narratives captions, single/multi-label prompts, respectively. $Cap_{rd}^-$, $Cap_{rl}^-$ and $Cap_{er}^-$ represents captions from random images, captions from relevant image, and true captions modified with errors. We can see the similarity scores from vision-language models can be easily distorted by hard positives and negatives.

| Model | $Cap^+$ | | | $Prompt_m^+$ | | | $Prompt_s^+$ | | | $Cap_{ln}^+$ | | |
|---|---|---|---|---|---|---|---|---|---|---|---|---|
| | $Cap_{rd}^-$ | $Cap_{rl}^-$ | $Cap_{er}^-$ | $Cap_{rd}^-$ | $Cap_{rl}^-$ | $Cap_{er}^-$ | $Cap_{rd}^-$ | $Cap_{rl}^-$ | $Cap_{er}^-$ | $Cap_{rd}^-$ | $Cap_{rl}^-$ | $Cap_{er}^-$ |
| CLIP-B | 94.78 | 82.77 | 28.10 | 74.39 | 50.12 | 7.19 | 55.69 | 30.17 | 4.47 | 81.29 | 60.00 | 13.57 |
| CLIP-L | 95.64 | 84.66 | 30.59 | 79.84 | 56.76 | 8.51 | 58.18 | 33.52 | 4.94 | 85.51 | 66.74 | 16.63 |
| OpenCLIP$_{B-400M}$ | 95.28 | 84.62 | 29.61 | 64.66 | 39.1 | 4.49 | 50.9 | 25.93 | 3.63 | 83.48 | 63.07 | 13.85 |
| OpenCLIP$_{B-2B}$ | 96.28 | 86.73 | 28.96 | 75.84 | 51.83 | 6.32 | 61.39 | 35.89 | 4.42 | 88.83 | 71.76 | 18.91 |
| OpenCLIP$_{L-2B}$ | 97.09 | 88.81 | 33.03 | 79.22 | 56.00 | 6.90 | 65.44 | 39.97 | 4.96 | 89.50 | 72.78 | 18.63 |
| OpenCLIP$_{H-2B}$ | 97.45 | 89.85 | 35.82 | 79.2 | 57.64 | 7.49 | 65.67 | 42.19 | 5.75 | 89.74 | 73.28 | 18.09 |
| UniCL$_{All}$ | 94.37 | 81.76 | 20.74 | 82.58 | 62.33 | 9.94 | **82.45** | **60.02** | 8.71 | 81.96 | 62.33 | 12.99 |
| KLITE | 92.47 | 77.67 | 16.45 | 75.71 | 53.60 | 9.03 | 69.98 | 47.06 | 8.47 | 79.81 | 59.24 | 11.16 |
| BLIP | 97.68 | 90.89 | 48.53 | 57.64 | 32.21 | 3.13 | 43.24 | 20.07 | 2.81 | 82.26 | 63.62 | 17.94 |
| BLIP$_{ft-coco}$ | 99.07 | 95.15 | **56.44** | 74.65 | 51.02 | 4.86 | 65.96 | 41.77 | 4.02 | 89.99 | 75.92 | 23.13 |
| BLIP$_{ft-coco-fusion}$ | **99.26** | **96.08** | 38.57 | 76.59 | 54.97 | 3.35 | 81.62 | 58.41 | 2.97 | 92.51 | 82.59 | 22.72 |
| FLAVA | 97.73 | 89.31 | 29.49 | **86.52** | **69.29** | **13.22** | 78.35 | 58.33 | **11.22** | **94.83** | **82.87** | **35.09** |
| NegCLIP | 96.6 | 87.37 | 51.88 | 65.32 | 39.91 | 6.7 | 61.32 | 34.52 | 6.09 | 76.70 | 53.93 | 13.33 |

pretraining data. In our study, we demonstrated that the specificity of text can distract vision-language scores that VLMs struggle to reflect the correctness faithfully.

**Evaluation protocol and dataset**   We use image-to-text retrieval as the proxy task to demonstrate that the scores of contrastive vision language models can easily be distracted. We build our experiments on images of the MSCOCO2017 dataset and their annotation of captions and bounding boxes. The setup of the image-to-text retrieval task is following. Given a query image and a set of positive and negative text, the score between the query image and each text is used for retrieving the positive text. Average Precision (AP) is the metric for evaluating the performance of each image and we report the mean Average Precision (mAP) of the whole dataset. Typically, the positive text are the captions annotated for the query images ($Cap^+$), and the negative text is the captions of other images in the data ($Cap_{rd}^-$). To test our hypothesis, we design the following hard positives and hard negatives.

- *Prompts of a single label* ($Prompt_s^+$): apply the classification prompts on one label of the query image. For example, "a photo of a dog".
- *Prompts of multiple labels* ($Prompt_m^+$): apply the classification prompts on all labels in the query image. For example, "a photo of dog, person, ball".
- *Captions from Localized narratives(Pont-Tuset et al., 2020)* ($Cap_l^+n$): the text descriptions that are much longer more informative than typical captions in MSCOCO and pretraining data.
- *Captions of relevant images* ($Cap_{rl}^-$): COCO captions of relevant images that have overlapping labels with the query image.
- *Captions with errors* ($Cap_{er}^-$): modifying true COCO captions of query images with errors by replacing a noun entity in the text with the name of a label that does not exit in the image. We use spaCy [1] for entity recognition.

The hard positives $Prompt_s^+$, $Prompt_m^+$ and $Cap_l^+n$ contain less or more information, although still correct, than the true captions $Cap^+$. They can examine if different specificity of the positive text can reduce the score. The hard negatives $Cap_{rl}^-$ and $Cap_{er}^-$ are similar to true captions but are wrong descriptions. They can examine whether the model can be robust to specificity and indicate the correctness of text input. Note that we use randomly chosen 100 negative texts for each query image for all image-text retrieval experiments and report results of CLIP ViT-B/32.

---

[1] https://spacy.io/

**Results and implications** We first plot a normalized histogram of visual-language scores between query images and various textual inputs in Figure 5. Figure 5-Left compares the scores from different types of positive texts. True COCO captions generate higher score than classification prompts (multiple-label prompts get higher scores than single-label prompts), and Localized-narratives who are overly-detailed captions surprisingly lead to lower scores than normal captions. The observations confirms our hypothesis that *the amount of information (specificity) in text can distort the scores* and the specificity that is closer to the training text leads to higher scores. Shown by Figure 5-Middle, among the negative text, captions from relevant images achieve slightly higher scores than random captions, while captions with modified errors achieve similarly high scores as the true captions and can strongly destroy the effectiveness of the score. The result verifies our hypothesis that *the similarity scores cannot distinguish the correctness*. When comparing the positive single-label prompts with different types of negative text in Figure 5-Right, single label positives $\text{Prompt}_s^+$ are even lower than hard negatives $\text{Cap}_{er}^-$, which is not desired.

Then, we report the image-to-text retrieval results in Table 3 when combining different positive and negative text. We can see that using harder positives or harder negatives can degrade image-to-text retrieval performance, and retrieving label prompts from captions with small errors is extremely hard. Comparing the performance of different models, we can see that the BLIP model with the fusion design fine-tuned on COCO is the best when the positive text are true captions which is nature since it is trained on the same data. however, results in worse performance when distinguishing poisoned captions. When the positive text is label prompts, FLAVA is the best or the second best model, probabally due to its additional uni-modal training data/loss. UniCL is the best when single-label prompts are the positives, which we think can be explained by the ImageNet21K classification dataset in its training data.

**Does fine-tuning on hard negatives solve the issues?** Similar to the fine-tuning strategy in Yuksekgonul et al. (2022), we can also adapt the proposed hard positive and hard negative generation method to augment the training data. However, this does not imply that the problem can be solved since the model will likely fail on cases not covered by our augmentation strategy. For exmaple, NegCLIP finetuned on their order-perturbated text still fail our benchmark or even worse than CLIP in some cases. We further study this and including more results in Appendix.

## 6 CONCLUSION AND DISCUSSION

With the increasing interest in applying vision-language models, we present a novel benchmark and comprehensive study on the behaviors that create challenges to be useful in the open-world settings. First, we demonstrate that VLMs perform inconsistently on concepts of different semantic granularity. Based on our experiments, the performance discrepancy is due to the biased distribution of training data. Second, we show that vision language scores mostly measure similarity rather than correctness and can be distracted by the specificity of text. The scores are higher when the specificity of text are closer to the captions in the training data. This issue cannot be systematically solved by fine-tuning wit hard text mining.

Although we do not propose new approaches to address found issues, there are directions where we can make efforts. First, the training data account for the granularity discrepancy and the incapability for correctness, and thus we can improve these issues by augmenting the text with a more balanced concept distribution and including hard negatives and positives to help the model to learn to output correctness instead of similarity. Hard negative training has been demonstrated to be effective for contrastive learning (Robinson et al., 2020) and VLMs (Yuksekgonul et al., 2022). Second, the two encoders + embedding similarity design naturally leads to difficulties in recognizing the correctness. For example, the true captions and the captions with small errors are supposed to have similar embedding from a uni-modal perspective, which leads to close scores with the same image embedding. Therefore, a more powerful cross-modality fusion module is necessary to reasoning between visual and textual features that enable opposite output on similar text input. Lastly, large language models (LLM) trained on significantly richer text data can potentially alleviate the challenges we observed. We designed a simple language-only experiment, included in Appendix, to demonstrate the potential of using generative LLM to address the observed challenges. Developing and evaluating VLMs with generative LLMs for recoginition tasks is interesting future directions.

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

## A  APPENDIX

## B  A TWO-LEVEL GRANULARITY BENCHMARK

In this section, we presents an simplified granularity benchmark with two-levels of semantic hierarchy. The results are consistent with our observations in the main paper.

**Two-level Dataset**   Our evaluation starts on a dataset with two levels of labels: $N_{cg}$ coarse-grained (CG) classes $Y_{cg} = \{y_{cg}^i\}$, where $i \in \{1, ..., N_{cg}\}$, and each CG class has $N_{fg}^i$ fine-grained (FG) children classes $Y_{fg}^i = \{y_{fg}^{i,j}\}$, where $j \in \{1, ..., N_{fg}^i\}$. In total, there are $N_{fg} = \sum_{i=1}^{N_{cg}} N_{fg}^i$ FG classes. To create our two-level classification dataset, we adapt the tiered-ImageNet Ren et al. (2018) benchmark, which has 608 FG classes (a subset of the original 1000 classes of ImageNet-1K) organized under 34 CG classes and covers 30,400 out of 50,000 ILSVRC-12 validation images.

**Evlauation protocol**   For *two-level* granularity, we measure the performance difference of CG classification between using direct predictions with CG prompts and propagated FG predictions. The simplest propagation method is to assign the predicted FG labels to their CG parents' labels. For instance, if an image is predicted as "golden retriever" in the FG classification, it is labeled with its CG parent class "animal." Intuitively, if a model exhibits consistent understanding of CG and FG concepts, the performance of CG classification using CG prompts should be similar to propagating the results from FG classification. An alternative way of propagating FG to CG concepts is using the aggregated embeddings of FG prompts for CG classifcation. Specifically, for the $i$-th CG class, we compute the average of the FG prompt embeddings as the CG prompt embedding:
$E_t^{prop}(y_{cg}^i) = \frac{1}{N_{fg}^i} \sum_{j=1}^{N_{fg}^i} E_t(y_{fg}^{i,j})$. We use top1 accuracy as the classification metric.

Table 4: Evaluating vision-Language model zero-shot classification performance (top-1 accuracy) on fine-grained classes (FG) and coarse-grained (CG) classes. The CG classification results are obtained through two methods: relating predicted FG class labels to their CG parents (CG_FG-label) and using the average of the FG prompt embeddings as the CG prompt embedding (CG_FG-emb). We measure the differences ($\Delta$) with CG classification using CG class prompts (CG_direct), which reveals the discrepancy in CG-FG performance of vision-language models.

| Model | Arch | Training data | FG_direct | CG_direct | CG_FG-label ($\Delta$) | CG_FG-emb ($\Delta$) |
|---|---|---|---|---|---|---|
| CLIP | ViT-B-32 | Private400M | 66.47 | 50.15 | 86.35 (+36.2) | 72.62 (+22.47) |
| Open-CLIP | ViT-B-32 | LAION400M | 63.82 | 35.98 | 84.08 (+48.1) | 69.65 (+33.67) |
| | ViT-B-32 | LAION2B | 69.78 | 45.54 | 87.39 (+41.85) | 71.54 (+26) |
| | ViT-L-14 | LAION2B | 77.72 | 49.74 | 91.83 (+42.09) | 76.49 (+26.75) |
| | VIT-H-14 | LAION2B | 80.39 | 52.22 | 92.86 (+40.64) | 77.43 (+25.21) |
| UniCL | Swin-B | YFCC14M | 41.14 | 37.37 | 69.67 (+32.3) | 59.75 (+22.38) |
| | Swin-B | IN21K | 30.6 | 53.14 | 66.26 (+13.12) | 59.5 (+6.36) |
| | Swin-B | IN21K+YFCC14M | 45.91 | 52.27 | 76.84 (+24.57) | 67.63 (+15.36) |
| | Swin-B | IN21K+YFCC14M+GCC15M | 60.17 | 51.9 | 83.44 (+31.54) | 68.37 (+16.47) |
| K-LITE | Swin-B | IN21K+YFCC14M+GCC15M | 54.75 | 44.92 | 81.85 (+36.93) | 71.05 (+26.13) |
| BLIP | ViT-B-16 | COCO+VG+CC+SBU +LAION+CapFilt-L | 55.41 | 42.09 | 80.92 (+38.83) | 69.69 (+27.6) |
| BLIP_ft-coco | | | 58.02 | 46.75 | 84.7 (+37.95) | 72.93 (+26.18) |
| FLAVA | ViT-B/16 | PMD70M | 59.48 | 50.11 | 83.37(+33.26) | 70.84 (+20.73) |

## C  A LANGUAGE ONLY STUDY

In the main paper, we have highlighted the issues faced by vision and language models (VLMs) in zero-shot recognition tasks, focusing on both granularity and correctness analyses. Since these analyses primarily involve working with different text inputs while keeping the visual inputs constant, improving the language encoder becomes a natural next step. We address the question of whether

language embeddings from pre-trained large-scale language models (LLMs) exhibit better behavior compared to VLMs. To investigate this, we design a language-only task.

Specifically, we conduct a text classification task that involves classifying fine-grained (FG) concepts to their corresponding coarse-grained (CG) concepts using the same two-level ImageNet dataset as in Section 4.1. This results in a 34-way classification task with 608 text samples (FG concept prompts). Similar to zero-shot image classification, we compute the cosine similarity between the language embeddings of FG and CG prompts and classify a FG concept to the CG concept with the highest similarity score. To incorporate the generative model GPT-3 for this task, we design the following zero-shot prompt:

> "Classify a given concept into one of the following classes: ${all coarse-grained concepts }.
> Q: ${a fine-grained_concept} A:"

Table 5 Presents the performance of LLMs[2] or the language encoder of VLMs on the language-only task. Surprisingly, LLMs, even when fine-tuned for sentence embedding, do not outperform the language encoder of VLMs. However, we find that GPT-3 performs significantly better in a generative manner. This suggests that when dealing with concept relationships on a larger scale where simple embedding similarity struggles, generative modeling may offer a more powerful approach to capture complex semantic knowledge and model the relationships effectively.

Table 5: Performance (accuracy) of classify a fine-grained concept to coarse-grain concept using language embedding models or generative language models.

| Model Type | FG-to-CG Text Classification Accuracy (%) |
| --- | --- |
| CLIP-B | 61.18 |
| OpenCLIP-L$_{\text{LAION2B}}$ | 55.76 |
| OpenCLIP-H$_{\text{LAION2B}}$ | 62.66 |
| UniCL | 52.96 |
| KLITE | 43.59 |
| BLIP | 50.00 |
| FLAVA | 57.40 |
| all-roberta-large-v1 | 51.81 |
| sentence-T5-large | 52.47 |
| sentence-T5-xl | 55.26 |
| GPT-3$_{\text{text-davinci-002}}$ | 71.17 |

## D  DOES FINETUING WITH HARD POSITIVE OR NEGATIVE TEXT HELP?

To investigate if vision-and-language models (VLMs) can accurately understand the correctness of text in relation to images, we generate hard positive and negative text samples. The question arises whether utilizing these hard text samples for training or fine-tuning VLMs would be beneficial. A recent study (Yuksekgonul et al., 2022) explores the understanding of compositional relationships in VLMs and proposes fine-tuning as a potential solution. Inspired by this, we fine-tune the CLIP-B model on MSCOCO training data using the hard positive (single/multi-label prompts) and negative text samples (captions from relevant images and true captions modified with errors) generated by our benchmarking strategy. We use the default finetuning hyperparameters in Yuksekgonul et al. (2022). For an ablation study, we also fine-tune CLIP-B without the use of hard samples. The performance of these models, along with NegCLIP from (Yuksekgonul et al., 2022), is reported in Table 6.

It is important to note that, in addition to hard text samples, NegCLIP utilizes hard image samples for contrastive learning, while our fine-tuning approach focuses solely on hard text samples. From the table, we can observe that fine-tuning on MSCOCO data without hard samples improves the

---

[2]We use pretrained models provided by `sentence-transformer https://github.com/UKPLab/sentence-transformers`

performance in distinguishing true captions from other negative captions. Both NegCLIP and our fine-tuned models further improve the performance on other hard-positive or negative retrieval tasks, albeit with a slight degradation in the performance of easier settings, such as $\text{Cap}^+$ vs $\text{Cap}^-_{rd}$ or $\text{Cap}^-_{rl}$.

Our fine-tuned model significantly outperforms NegCLIP in challenging settings due to a better alignment between the training and testing text samples. However, even with the improvements, NegCLIP and our fine-tuned model still struggle with difficult settings, such as single/multi-label prompts $\text{Prompt}^+_s$ and $\text{Prompt}^+_m$ vs $\text{Cap}^-_{rl}$ which are poisoned captions with minor modifications. In $\text{Cap}^+_{ln}$ vs $\text{Cap}^-_{rd}$ and $\text{Cap}^-_{rl}$, all fine-tuned models are getting worse than original CLIP. These results highlight the limitations of relying solely on fine-tuning with hard samples, as the models are likely to fail on cases not covered by our augmentation strategy. Therefore, while fine-tuning with hard samples can alleviate some of the observed issues in our study, it may not provide a complete solution, particularly when considering the challenges of scaling up hard sample generation to encompass a wide range of possible cases. A more systematic solution is urgently needed.

Table 6: The performance of image-to-text retrieval task on MSCOCO-2017 measured by mean Average Precision (mAP). We compare various fine-tuned CLIP-B models. Different columns represent different choices of positive and negative texts. $\text{Cap}^+$ ,$\text{Cap}^+$ ,$\text{Cap}^+_{ln}$, $\text{Prompt}^+_s$ and $\text{Prompt}^+_m$ stands for true COCO captions, Localized-narratives captions, single/multi-label prompts, respectively. $\text{Cap}^-_{rd}$, $\text{Cap}^-_{rl}$ and $\text{Cap}^-_{er}$ represents captions from random images, captions from relevant image, and true captions modified with errors.

| Model | $\text{Cap}^+$ | | | $\text{Prompt}^+_m$ | | | $\text{Prompt}^+_s$ | | | $\text{Cap}^+_{ln}$ | | |
|---|---|---|---|---|---|---|---|---|---|---|---|---|
| | $\text{Cap}^-_{rd}$ | $\text{Cap}^-_{rl}$ | $\text{Cap}^-_{er}$ | $\text{Cap}^-_{rd}$ | $\text{Cap}^-_{rl}$ | $\text{Cap}^-_{er}$ | $\text{Cap}^-_{rd}$ | $\text{Cap}^-_{rl}$ | $\text{Cap}^-_{er}$ | $\text{Cap}^-_{rd}$ | $\text{Cap}^-_{rl}$ | $\text{Cap}^-_{er}$ |
| CLIP-B | 94.78 | 82.77 | 28.1 | 74.39 | 50.12 | 7.19 | 55.69 | 30.17 | 4.47 | **81.29** | **60.00** | 13.57 |
| CLIP-B$_\text{ft-coco}$ | **96.98** | **88.15** | 35.04 | 71.65 | 47.16 | 5.41 | 60.4 | 33.53 | 4.09 | 80.52 | 57.96 | 10.66 |
| NegCLIP | 96.6 | 87.37 | 51.88 | 65.32 | 39.91 | 6.7 | 61.32 | 34.52 | 6.09 | 76.70 | 53.93 | 13.33 |
| Ours | 96.21 | 85.9 | **75.74** | **93.37** | **75.46** | **55.95** | **83.22** | **57.98** | **29.05** | 78.54 | 55.11 | **31.74** |

# E    LIMITATIONS OF OUR STUDY

While our study provides valuable insights into the challenges and limitations of vision-and-language models (VLMs) for zero-shot visual recognition, it is important to acknowledge several limitations. Firstly, our experiments primarily focus on a specific set of VLMs, datasets, and evaluation metrics. While we have made efforts to select representative models and datasets, our findings may not fully generalize to the entire landscape of vision and language models. Generalizing the results to other VLM architectures or datasets requires further investigation and experimentation.

Secondly, our study is conducted within the context of the evaluation protocols and benchmarks we have proposed. While we have designed these protocols to address the challenges of zero-shot recognition in open-world settings, it is important to recognize that these benchmarks may not fully capture the complexities and variations present in real-world scenarios. Real-world applications may involve different types of data, varied distributions, and additional challenges that are not fully accounted for in our study.

Furthermore, the scalability of hard sample generation, as used in our fine-tuning experiments, presents a practical limitation. Generating diverse and representative hard positive and negative samples can be computationally expensive and time-consuming. Scaling up the generation process to cover a wide range of positive and negative cases with diverse variations poses a significant challenge and may require more efficient and scalable methods.

