# OpenReview forum: "Benchmarking Zero-Shot Recognition with Vision-Language Models: Challenges on Granularity and Specificity"
_ICLR.cc/2024/Conference — Submitted to ICLR 2024_

### Official Review · Reviewer_jHnf · 2023-10-31

**Soundness:** 3 good
**Presentation:** 4 excellent
**Contribution:** 4 excellent
**Rating:** 8
**Confidence:** 5

**Summary:**

This paper focuses on exploring the zero-shot recognition of Vision-Language Models (VLMs) in terms of Granularity and Specificity. Driven by this objective, this paper provides a benchmark to test the behavior of VLMs on different levels of granularity.

The main two observations are  1) VLMs cannot correctly classify images from both fine-grained and coarse-grained perspectives. They are better at moderately fine-grained concepts.  2) In the meantime, this work also shows VLMs cannot detect whether the text correctly describes the given image. Instead, they could be sensitive to the specificity of text and struggle to distinguish between the challenging positives like single-label prompts, and hard negatives like poisoned captions with small changes.

**Strengths:**

+ [*The motivation is sound and convincing*] I like the idea of studying the VLM from granularity and specificity. For each perspective, this work provides a corresponding benchmark for measuring the performance. Such a study can provide insights into the zero-shot performance of VLM.

+ [The evaluation way is interesting and reasonable] The evaluation protocol compares a) the prompt of the coarse-grained class and b) aggregates the predictions from prompts of its fine-grained children classes. This is intuitive and makes sense to me. From such protocol, this work draws the claim that VLMs are better at moderately fine-grained concepts. But I do have a concern, please see the weakness.

+ [Specificity Robustness is well-delivered] From such analysis, we know the scores of contrastive vision language models can easily be distracted. I could guess such observation may be caused by the training data where the text is uncorrupted. I am wondering any solution to mitigate such specificity.

**Weaknesses:**

- [Hierarchical Labels vs. Coarse-grained Labels] There is a paper (Chils: Zero-shot image classification with hierarchical label sets in ICML 2023) that shows that using fine-grained/ subclass-level labels helps zero-shot classification. It seems the conclusions of this work and ICML work are different. Please discuss this.

- [Specificity of Text] I think the observation that VLMs could be sensitive to the specificity of text is straightforward. When training VLMs, they are given specific texts. If we use the proposed modified version for training, I would expect them to perform better. Please comment on my thoughts.

***Post Rebuttal***

Thanks for the reponse, which is helpful to adress my intial concerns. Please make sure you include these disucssions in the revised version. I maintain my recomendation of Accept.

**Questions:**

- Please discuss the different observations between your work and ICML work. Also, comment on my thoughts for the bias caused by the training manner.

- Section 4.2 (Impact of Pre-training Data) is not clear to me. Please elaborate your points here.

---

> ### Author Response · Authors · 2023-11-21
> **Response to Reviewer jHnf**
>
> Dear Reviewer jHnf,
>
> We appreciate your feedback and are thankful for your recognition of the value and novelty of our work. We would like to address your concerns and questions below.
>
> * The differences between our observations and the CHiLS [1] paper as follows:
> CHiLS demonstrates that mapping predictions from leaf classes across granularity levels 1-6 can enhance results for ImageNet classification (Table 1 in [1]), aligning with our findings in a multi-label classification context (Figure 3 in our paper). However, our analysis extends to all levels of the ImageNet hierarchy, up to 16. We observed distinct behaviors when the leaf classes are extremely fine-grained. In such cases, propagating their predictions to parent classes may not be beneficial. For instance, certain classes at level 16 exhibited negative changes when predictions were propagated from children at level 17, as illustrated in our Figure 3-Right.
>
> * Regarding the potential improvement in Specificity through modified training, our Appendix (Section C) explores this. We finetuned CLIP-B on COCO with diverse text samples generated by our benchmarking strategy. While fine-tuning with challenging text samples shows promise, it falls short of a comprehensive solution. Notably, the models struggle with cases outside the augmentation strategy's scope, underscoring the limitations of relying solely on fine-tuning with hard samples. Moreover, achieving certain augmentation goals, like enhancing alt-text to increase correct information, presents significant challenges.
>
> * To clarify Section 4.2 (Impact of Pre-training Data), we compared the performance of UniCL models trained on distinct datasets. These datasets include ImageNet21K, featuring classification prompts (e.g., “a photo of {}”) and class names, and image alt-text data (YFCC-14M and GCC-15M).
>
>   * UniCL trained solely on image alt-text data (UniCL_YFCC) outperforms UniCLIN21K in FG leaf label classification (35.75 vs. 26.28 mAP).
>   * UniCL_IN21K, trained on ImageNet21K, excels in CG ancestor raw performance due to the inclusion of all CG classes in IN21K, surpassing CLIP and OpenCLIP trained on larger-scale alt-text data.
>   * Training UniCL on a combination of IN21K with Alt-text Data (YFCC14M and GCC15M) improves FG classification but slightly worsens CG classification, leading to a larger CG-FG discrepancy.
>   * Scaling up alt-text training data: Larger models, such as OpenCLIP-H-2B trained on LAION2B, show inferior CG ancestor raw performance and a more pronounced CG-FG discrepancy than UniCL_IN21K, indicating that merely scaling up the training data or model sizes is insufficient to resolve this discrepancy effectively.
>
>   We will modify our draft to make this section clearer.

---

### Official Review · Reviewer_stdj · 2023-10-31

**Soundness:** 2 fair
**Presentation:** 3 good
**Contribution:** 2 fair
**Rating:** 3
**Confidence:** 4

**Summary:**

The paper presents a benchmark for evaluating the granularity and specificity of text prompts in vision-language pretraining such as CLIP. Experiments are conducted over several state-of-the-art vision language models (e.g., CLIP, OpenCLIP-LAION, BLIP, etc.). For granularity, the benchmark includes images from ImageNet-1K dataset in which the label hierarchy is extracted by WordNet. The evaluation includes matching the cosine similarity of an object name with its parent and child category names. Results show that the models are better at recognizing moderately fine-grained concepts than course-grained concepts. For specificity, the data for evaluation includes captioning example from MS-CoCo, in which caption text is adjusted. The results show that short (single label) or long captions can produce lower scores than captions with correct specific details that also include small errors. Overall, the evaluation points to limitations of these models in the zero-shot classification task.

**Strengths:**

The study and finding of the limitations of VLMs is of interest.

**Weaknesses:**

The evaluation on the technical side is rather straightforward, and I am not sure how surprising the results are. Although this study may provide few valuable insights, I do not think it meets the ICLR novelty threshold.

**Questions:**

none

---

> ### Author Response · Authors · 2023-11-22
> **Response to Reviewer stdj**
>
> Dear Reviewer stdj,
>
> Thank you for your time for reviewing our submission.
>
> We acknowledge your concern regarding the straightforward nature of our technical evaluation. However, we wish to emphasize that the straightforwardness of our approach was crucial for ensuring the clarity and replicability of our results. Our methodology, while seemingly intuitive, was designed to rigorously test VLMs in scenarios that have not been explicitly showcased or examined in existing literature.
>
> Regarding the novelty of our work, we believe our contribution is significant in several aspects:
> Empirical Evidence and Novel Benchmarks: We have developed new benchmarks and conducted comprehensive analyses, providing empirical evidence of the VLMs' performance across different granularity and specificity levels.
> Development of a New Evaluation Protocol: A major novelty of our research lies in the creation of a new evaluation protocol and the modification of existing datasets to fit this protocol. This effort was non-trivial and has not been previously undertaken to our knowledge. We would greatly appreciate it if the reviewer could provide references to previous works that are dedicated in the same topic of our work.
> We also provide deep analysis on how the training data impact the observed granularity issue and if finetuning can solve the specificity issue.
>
> We believe that our research fills a crucial gap in the current understanding of VLMs, and we hope that our responses clarify the novelty and depth of our contributions to this important area of study.

---

### Official Review · Reviewer_pPRh · 2023-11-01

**Soundness:** 2 fair
**Presentation:** 2 fair
**Contribution:** 2 fair
**Rating:** 3
**Confidence:** 4

**Summary:**

Authors analyze the performance of off-the-shelf VLMs (e.g. CLIP, Open-CLIP, BLIP, FLAVA, etc.) on their ability to correctly match paired visual and textual descriptions at different levels of granularity and specificity. To this end, authors propose two benchmarking protocols based on ImageNet and COCO-Captions. They find that VLMs achieve the highest accuracy at recognizing "moderately fine-grained concepts" and VLMs struggle to correctly rank different captions with incorrect details. Authors evaluate several VLMs under their proposed setup and provide analysis on their results.

**Strengths:**

- Representative Baselines. Authors evaluate a representative set of VLMs with different backbones and pre-training datasets.
- Sensible Analysis. Authors conduct sensible analysis on the OTS pre-trained models to back up their claims. All of the results of the analysis are intuitive and reasonable.

**Weaknesses:**

- Lack of Novel Insights. The primary claims in this paper about the performance of VLMs on specificity and granularity are not new, primarily because these have been explored in the language community [1, 2]. This is particularly important because both specificity and granularity are primarily language-based reasoning tasks. For example, an image of a leopard is unlikely to be close in feature space to an image of a bird even though both are examples of animals. In contrast, the language embedding contributes significantly more to this problem. Similarly, the evaluation of specificity is largely dependent on the text encoder's ability to capture fine-grained details (which is well studied in the language community.) Further, the conclusion that the distribution of data plays a significant role in VLM performance has been explored in [1]. Since VLMs are often trained on uncurated web-sources, we expect that they will capture the distribution and correlations found on the web. Therefore, the conclusion that VLMs are better at recognizing "moderately fine-grained concepts", is simply a restatement that VLMs are better at matching images with commonly occurring captions. To take an extreme example, a VLM is unlikely to score the concept "object" and an image of a "leopard" highly since these hierarchical correlations (while true) rarely occur in the training data. Similarly, a VLM is unlikely to score the concept of "leopard" highly with its scientific name "Panthera pardus" because it is also rarely seen during training. With respect to the specificity benchmark, the insight that vision language models can be easily distracted has also been explored in [2] (Note that although [2] focuses on language, the textual bias of the specificity task makes this work relevant).
- Textual Bias in Specificity Benchmark. As the authors point out in the supplement, language-only methods are competitive on this benchmark, suggesting that the protocol should be amended to more accurately to evaluate vision-language capabilities, rather than just language understanding. In fact, many VLM benchmarks face the same problem of not requiring vision to solve the task.
- On the Importance of Granularity from Vision. The granularity benchmark actually tests a model's alignment with word-net, which may not be universally accepted. For example, searching the web for "road-cone" and "barrier" show two visually distinct items. Although people may agree that road-cone and barrier are related, it is unclear if road-cone is a child class of barrier. Therefore, simply training a VLM in alignment with word-net rather than the broader internet may artificially boost performance on this benchmark, but perform worse in practice.

References

[1] Large Language Models Struggle to Learn Long-Tail Knowledge. Kandpal et. al. ICML 2023

[2] Can Large Language Models Truly Understand Prompts? A Case Study with Negated Prompts. Jang et. al. NeurIPS Workshop 2022.

[3] When and Why Vision-Language Models Behave like Bags-Of-Words, and What to Do About It? Yuksekgonul et. al. ICLR 2023.

**Questions:**

- What is the Summary Metric? Although the in-depth analysis is appreciated, it is difficult to identify the best and worst performing methods (as is common in a benchmark). How can the methods be ranked for both benchmarks?
- How Should Training be Amended? In light of the analysis of this paper, what are the right steps to amend VLM pre-training. Although authors suggest fine-tuning with hard-negative text prompts, this again seems to bias the model to this particular benchmark (no longer benchmarking zero-shot recognition), and does not provide a solution in general.

Please revise the text of this paper, there are many spelling and grammar errors (e.g. Figure 5 poisoneded)

---

> ### Author Response · Authors · 2023-11-22
> **Response to Reviewer pPRh**
>
> Dear Reviewer pPRh,
>
> Thank you for your valuable feedback on our submission. We've carefully considered the concerns and questions you've raised and would like to offer the following responses.
>
> - **Lack of Novel Insights:**
>   - Although some issues relevant to detailed language understanding may have been studied in the NLP community, our research brings a novel perspective by investigating these issues in the context of VLMs. Unlike LMs that process only textual content, VLMs require an intricate understanding of how visual elements correspond with textual descriptions with unique challenges. For instance, understanding the granularity in the context of VLMs involves not only textual recognition of 'leopard' vs 'animal' but also the correct visual identification of these categories in diverse images.
>   - While we acknowledge the relevance you've drawn with studies [1] and [2], we would like to underscore the distinct contributions of our research in the realm of Vision-Language Models (VLMs). [1] primarily examines the memorization and retrieval capabilities of LMs in QA tasks in relation to the frequency of information in training datasets. In contrast, our work evaluates VLMs' ability to interpret and relate visual concepts with varying levels of granularity and text of various specificity, a challenge that inherently involves interpreting visual stimuli alongside textual data. Reference [2] ("Can Large Language Models Truly Understand Prompts?") focuses on the comprehension of negated prompts in LMs. Our research on specificity, however, is concerned with how the VL matching score can be distorted by the amount of information in the text, a distinct and unexplored area in the field of VLMs.
>   - We acknowledge the intuitive aspect of our finding that "VLMs are better at matching images with commonly occurring captions." However, our research adds significant value by grounding this intuition within the newly formulated benchmarks for granularity and specificity evaluation. This framing provides a structured and empirical approach to understanding how VLMs handle varying levels of granularity and specificity  in the complex interplay of visual and textual data, offering insights that extend beyond the general understanding of VLMs' capabilities based on their training data distribution.
>
> - **Textual Bias in Specificity Benchmark**:
>   - We would like to clarify a critical aspect of our study that seems to have been misunderstood. Our language-only study was not aimed at demonstrating the competitiveness of language-only methods on the VLM benchmark. Instead, its purpose was to investigate whether the granularity issue observed in VLMs also manifests in language modeling alone.
>   - We specifically designed a text-classification task based on two-level granularity concepts, derived from our VLM benchmark, to assess if language models exhibit similar granularity issues. Our findings revealed that while text encoders in VLMs and embedding-type LMs struggle with this task, larger generative Language Models (LLMs), such as GPT-3.5, show better performance. However, it remains unclear whether this improved performance is attributable to the generative modeling approach or simply the larger scale of these models.
>
> - **On the Importance of Granularity from Vision**
> WordNet for its structured and widely-accepted semantic relationships, which provide a clear framework for evaluating VLMs' understanding of granularity. We believe overall it defines the most widely accepted conceptual relations. To clarify, the example “road-cone” and “barrier” we used to explain the granularity issue is not part of WordNet. We believe WordNet’s semantic relationships provide a solid foundation for evaluating VLMs' understanding of granularity while it may not capture every nuance of real-world visual categories. We will include the discussion in the limitation section.
> - **Summary Metric:** our study does not employ a single summary metric as it evaluates VLMs across two separate benchmarks: granularity and specificity, each testing different aspects of VLM capabilities. Consequently, each benchmark has its specific set of metrics that appropriately capture the distinct properties we aim to assess. This approach ensures a more nuanced and accurate evaluation of VLM performance in diverse scenarios.
> - **How Should Training be Amended?**
> We acknowledge that altering the training regime to fit a specific benchmark can introduce biases. Using LLM to augment more diverse text examples may improve it. We also recognize that this is not a universal solution to the inherent challenges in embedding matching based models. Our language-only study indicates that generative models, such as GPT-3.5, show promising results. Therefore, exploring the use of generative VLMs might be a fruitful direction for future research, despite the trade-off in computational cost.
>
> We will carefully review our draft to correct any spelling or grammatical errors.

---

### Official Review · Reviewer_PhzX · 2023-11-01

**Soundness:** 3 good
**Presentation:** 3 good
**Contribution:** 2 fair
**Rating:** 6
**Confidence:** 4

**Summary:**

This paper presents new benchmarks for Vision-Language Models (VLMs) in zero-shot recognition tasks, emphasizing granularity and specificity. The authors use adapted ImageNet and MS-COCO datasets to test VLMs' performance across varied granularity levels and their sensitivity to text specificity. Key findings reveal that VLMs excel at recognizing moderately fine-grained concepts but struggle with high-level coarse concepts and are sensitive to the specificity of language prompts. This in-depth evaluation uncovers significant limitations in state-of-the-art models like CLIP.

**Strengths:**

1. The paper is clear, and well-written.
2. The authors point out an interesting problem regarding granularity and specificity existing in current vision-language alignment models. It designed two simple benchmarks to evaluate those perspectives.  Their benchmarks address an important gap, making the paper a valuable resource for other researchers.
3. One of the standout findings, the improved performance through direct propagation from leaf nodes, offers fresh perspectives in the domain of zero-shot recognition. This insight could influence future model designs and strategies.

**Weaknesses:**

1. Lack of in-depth Analysis: The influence of prompt design on classification is a critical aspect that seems to be overlooked in the paper. An in-depth analysis of how prompt design effectiveness may sway classification results, especially in terms of granularity, is necessary to provide a comprehensive evaluation.

2. Going Beyond Common Knowledge: The paper highlights that performance in granularity and specificity improves when the testing scenario mirrors the training set, a well-known fact in the field. A more substantial contribution could be made by delving deeper into this issue, perhaps by providing baseline solutions or strategies to enhance granularity and specificity in pre-trained models.

3. Connection to Specific Vision Tasks: While the paper provides a high-level motivation for the need of such benchmarks, it stops short of connecting the dots to specific vision tasks. A more robust argument could be made by demonstrating how improvements in the proposed benchmarks translate to advancements in other vision-language tasks, such as open vocabulary object detection[1], open vocabulary tracking[2], and text-to-image generation[3]. At least, a high-level discussion is needed in this regard.

These points aim to encourage a more thorough exploration of the topics and a clearer demonstration of the benchmarks’ applicability to real-world tasks.

[1] Gu, Xiuye, et al. "Open-vocabulary object detection via vision and language knowledge distillation." arXiv preprint arXiv:2104.13921 (2021).

[2] Li, Siyuan, et al. "OVTrack: Open-Vocabulary Multiple Object Tracking." Proceedings of the IEEE/CVF Conference on Computer Vision and Pattern Recognition. 2023.

[3] Rombach, Robin, et al. "High-resolution image synthesis with latent diffusion models." Proceedings of the IEEE/CVF conference on computer vision and pattern recognition. 2022.

**Questions:**

Evaluation Using COCO: Why are COCO's original captions more effective than detailed ones? Is there any assurance that the model hasn't been exposed to any COCO images or captions during training?

---

> ### Author Response · Authors · 2023-11-21
> **Response to Reviewer PhzX**
>
> Dear Reviewer PhzX,
>
> We appreciate your thorough review of our paper and insightful comments. Below, we address each of your concerns:
>
> - **The influence of prompt design**:
>
>   We appreciate your comments on the influence of prompt design in classification, particularly regarding granularity. Our study utilized popular prompt templates from the CLIP paper, considering its broad application in vision-language (VL) model research. To further explore this aspect, we conducted additional analyses using a simplified prompt format: “a picture of {}.” Our updated results, which are in the table below, affirm that our conclusions on granularity issues remain consistent under this varied prompt design approach.
>
>   | Model                     | Leaves | Ancestor$_{raw}$ | Ancestor$_{child}$ | Ancestor$_{leaf}$ |
>   |---------------------------|--------|--------------|----------------|---------------|
>   | CLIP ViT-B/32             |  48.17 |        22.96 | 43.01(+20.05)  | 55.93(+32.97) |
>   | CLIP ViT-L/14             |  60.77 |        29.73 | 52.98(+23.25)  | 67.13(+37.40) |
>   | OpenCLIP B-400m           |  50.93 |        22.07 | 43.99(+21.92)  | 58.66(+36.59) |
>   | OpenCLIP B-2B             |  57.04 |         25.8 | 49.20(+23.40)  | 64.27(+38.47) |
>   | OpenCLIP ViT-L-14 laion2b |  67.42 |        31.98 | 57.76(+25.79)  | 74.16(+42.19) |
>   | OpenCLIP ViT-H-14 laion2b |  68.87 |        32.19 | 58.44(+26.25)  | 74.64(+42.45) |
>   | UniCL_YFCC                |  30.73 |        16.26 | 30.99(+14.73)  | 42.06(+25.80) |
>   | UniCL_IN                  |  22.84 |        34.93 | 35.56(+0.63)   | 35.53(+0.60)  |
>   | UniCL_YFCC+IN             |  40.92 |        32.09 | 45.34(+13.25)  | 54.27(+22.18) |
>   | UniCL_all                 |  44.43 |        28.78 | 44.13(+15.35)  | 55.33(+26.55) |
>   | KLITE                     |  45.82 |        26.93 | 45.26(+18.33)  | 58.03(+31.10) |
>   | BLIP                      |  44.87 |        21.14 | 41.21(+20.07)  | 54.45(+33.31) |
>   | BLIP_ft                   |  45.88 |        23.62 | 43.99(+20.37)  | 57.12(+33.51) |
>   | FLAVA                     |  36.05 |        17.57 | 33.99(+16.42)  | 45.68(+28.10) |
>
> - **Going Beyond Common Knowledge:**
>   We understand your concern that our findings about performance improvement under training-like scenarios may appear intuitive. Nevertheless, our contribution lies in the empirical evidence and systematic exploration of this phenomenon, which has not been explicitly explored in previous research. Our benchmarks provide a foundation for future in-depth studies in this area. Furthermore, we explored the efficacy of simple fine-tuning with hard examples to address the specificity issue. Detailed in Appendix (Section C), we fine-tuned the CLIP-B model on COCO using various text samples generated through our benchmark strategy. While fine-tuning shows some promise, it falls short as a comprehensive solution, particularly in instances outside the augmented sample scope. This finding underscores the limitations inherent in relying solely on fine-tuning with hard samples.
>
> - **Connection to Specific Vision Tasks:**
>
>   We are grateful for your suggestion to link our benchmarks to specific vision-language tasks. Downstream VL tasks like open-vocabulary detection/tracking and text-to-image generation often leverage pre-trained VL models assessed in our benchmarks. They often leverage the pretrained VL models tested in our benchmarking by knowledge distillation or apply similar image-text embedding alignment strategies on task-specific models with comparable image-text data. Consequently, they are subject to the same granularity and specificity issues we identified. Our improvements in these benchmarks should, therefore, translate to advancements in these tasks, yielding more robust pre-trained VL models and improved training methodologies. We will incorporate this discussion into our manuscript to underscore our benchmarks' practical implications in enhancing VLM performance across various applications.
>
> - **Regarding your question on Evaluation Using COCO:**
>   We believe the original captions are more effective due to their similarity in information content with the alt-text in VL training data, which is typically shorter than Localized Narratives captions. While there is overlap between COCO images and web-crawled datasets like LAION and YFCC, COCO's distinct annotation process separates its captions from these datasets. Recognizing your concern about potential model exposure to COCO images and captions during training, we note that such exposure is a common challenge in VL model evaluations. However, given the vast scale of pretraining data (billions of image-text pairs) compared to the size of the COCO dataset (hundreds of thousands), the models' behavior is predominantly shaped by the broader training data spectrum, where COCO's portion is minimal.

---

### Official Review · Reviewer_1Zb8 · 2023-11-03

**Soundness:** 4 excellent
**Presentation:** 3 good
**Contribution:** 3 good
**Rating:** 6
**Confidence:** 4

**Summary:**

The paper proposes to benchmark the zero-shot recognition ability of VLM in the axes of varying granularity levels and the specificity of language inputs. The findings show that existing VLMs tend to struggle with semantic granularity and are sensitive to text specificity. The study can help guide future works to benchmark and improve VLMs in these axes, making VLMs more robust and useful at scale.

**Strengths:**

The paper presents a thorough study on the zero-shot recognition capability of a suite of VLMs in terms of granularity and specificity. Both are important areas that the community should be aware of.

The findings on granularity are informative as it shows the VLMs are best at recognizing concepts in similar granularity as the training data distribution (raw scores of ancestor are much worse than aggregating max scores of the leaves). Figure 4 (left) connects the empirical findings of Figure 3 to the training data, which is convincing.

The observation of specificity is quite informative as it shows the model is best at associating images and texts with the right amount of information.

The studies in Table 2 and 3 are performed with a broad range of VLM, which makes the conclusion robust to modeling choice.

**Weaknesses:**

1. Apart from the analyses, it'd be great to explore some simple ideas to mitigate the granularity and specificity issues in these VLMs by light-weight finetuning. Some ideas include:
- Augment the alt-text with LLM to increase the amount of information
- Generate hard-negatives by replacing the nouns
- Adjust the granularity by replacing the fine-grained concepts with their parent in the hierarchy
Since the analysis points to the mismatch between training and test data distribution, it'd be natural to bridge the distribution gap by data augmentation.

2. It's not obvious how to translate the findings of this work into improvements in the downstream application of CLIP. Take the granularity study for example. Let's say we want to build an open-vocabulary detector on the some super categories of LVIS. To achieve the best performance, we'd need a way to generate the fine-grained categories associated with those super-categories, where the super-categories may not be in the WordNet hierarchy. One idea is to use the LLMs to help generate fine-grained categories so that we can apply the max-score aggregation in Fig 2 (c).

3. Although the analyses are comprehensive and interesting, the findings are not very surprising. It seems to boil down to the limitations of the data that naturally exists at large-scale at the end of the day. I'd recommend to take the discussion section outside of conclusion and expand on it more to address the limitations.

**Questions:**

See weakness.

---

> ### Author Response · Authors · 2023-11-21
> **Response to Reviewer 1Zb8**
>
> Dear Reviewer 1Zb8,
>
> Thank you for your detailed and insightful review of our submission. We address your questions as follows:
>
> - **Explore Simple Ideas to Mitigate Granularity and Specificity Issues by Lightweight Fine-tuning**:
>
>   We appreciate your suggestion to explore lightweight fine-tuning methods. Our appendix (Section C) investigates this for the specificity issue, where we finetuned CLIP-B on COCO using various text samples generated by our benchmarking strategy. While fine-tuning with hard text samples shows some promise, it’s not a comprehensive solution. In particular, we found that the models tend to fail on cases not covered by the augmentation strategy. This highlights the inherent limitations of relying solely on fine-tuning with hard samples. It’s also important to note that some augmentation goals, such as "augmenting the alt-text with LLM to increase the amount of information", can be challenging to achieve. There is no guarantee that the LLM-generated information will be well aligned with the image, posing a significant challenge for ensuring model accuracy and relevance. We concur that the ideas you have suggested are intriguing and will discuss them as potential future directions, acknowledging the complexities involved.
>
>
> - **Translating Findings to Downstream Applications:**
>
>   Our study provides essential insights into the limitations of current VLMs in terms of granularity and specificity, forming a basis for enhancing these models for practical applications such as open-vocabulary object detection. We acknowledge the potential of applying our findings to practical applications, as you kindly suggested, and will include a discussion in our revised draft on this promising yet extensive topic, which requires further investigation beyond our current scope.
>
>
> - **Regarding the "Not Surprising" Nature of Findings:**
>
>   We believe that the seemingly intuitive nature of findings does not diminish their importance. Our work, like recent studies on VLMs' compositional understanding limitations, provides vital empirical evidence, despite some findings being in line with expectations. These insights are crucial for advancing the field. We have discussed the limitations in Appendix Section E and will expand this discussion to further articulate the significance of our findings.

---

### Author Response · Authors · 2023-11-22
**Rebuttal Summary**

We would like to express our gratitude to the reviewers for their insightful feedback on our paper. We sincerely appreciate the reviewers' recognition of the strengths of our work:

**Significance**(1Zb8, PhzX, stdj, jHnf) Our study on the zero-shot recognition capability of VLMs in terms of granularity and specificity is important and interesting. The findings address an important gap making the paper a valuable resource for other researchers.

**Technique Soundness** (1Zb8, pPRh, jHnf, pPRh): The evaluation is reasonable and thorough, covering a representative set of VLMs. The analysis is convincing and backup our claims. The findings are informative.

**Quality of Writing** (PhzX): The paper is clear, and well-written.


We also thank the reviewers for their questions and constructive suggestions for further improving our work. We have replied to each reviewer separately to address the concerns and questions of each reviewer and summarize the major ones here:

* **Novelty and "Not Surprisingly" Nature of Findings:**

  Our investigation into the limitations of Vision-Language Models (VLMs), particularly in terms of granularity and specificity, provide novel insights. A significant aspect of our work is the development of a new evaluation protocol, specifically tailored to assess these facets and the modification of existing datasets to fit this protocol – a non-trivial effort. While some of our findings might initially appear intuitive, they provide empirical evidence that has not been previously comprehensively demonstrated in existing literature. Our benchmarks set a solid foundation for future in-depth studies in this area. Additionally, our explorations on fine-tuning and the language-only study, as detailed in the Appendix, contribute further insights for prospective research.

* **Differentiation from the Studies of LMs on Language Understanding in NLP Literatures**

  We acknowledge the similarities identified by the reviewers between our work and NLP studies. However, our research distinctly stands apart. Unlike LMs that process solely textual content, VLMs necessitate a nuanced understanding of the interplay between visual elements and textual descriptions, posing unique challenges. For instance, the comprehension of granularity within the context of VLMs involves not just the textual recognition of terms like 'leopard' versus 'animal', but also their accurate visual identification across varied images. Additionally, our focus on granularity and specificity introduces a different formulation compared to the long-tail knowledge QA [1] and negated prompts [2] explored in referenced NLP studies.

  [1] Large Language Models Struggle to Learn Long-Tail Knowledge. Kandpal et. al. ICML 2023

  [2] Can Large Language Models Truly Understand Prompts? A Case Study with Negated Prompts. Jang et. al. NeurIPS Workshop 2022.

* **Connection to Specific Tasks and Practical Applications**:

  Our benchmarks are closely linked to various Vision-Language tasks, such as open-vocabulary detection and text-to-image generation. The insights and enhancements derived from our research have significant practical implications, laying the groundwork for improved performance in these tasks. This is particularly relevant considering that these tasks often build upon the pre-trained VLMs studied in our work or employ similar approaches and datasets for training.

---

> ### Author Response · Authors · 2023-11-22
> **Rebuttal Summary continued**
>
> * **Additional Experiment** - As the reviewer suggested, we further studied the impact of prompt design on classification for granularity study. We conducted supplementary experiments using a simplified prompt  “a picture of {}.”. The results reconfirm our initial findings regarding granularity issues in VLMs and further solidify our conclusions.
>   | Model                     | Leaves | Ancestor$_{raw}$ | Ancestor$_{child}$ | Ancestor$_{leaf}$ |
>   |---------------------------|--------|--------------|----------------|---------------|
>   | CLIP ViT-B/32             |  48.17 |        22.96 | 43.01(+20.05)  | 55.93(+32.97) |
>   | CLIP ViT-L/14             |  60.77 |        29.73 | 52.98(+23.25)  | 67.13(+37.40) |
>   | OpenCLIP B-400m           |  50.93 |        22.07 | 43.99(+21.92)  | 58.66(+36.59) |
>   | OpenCLIP B-2B             |  57.04 |         25.8 | 49.20(+23.40)  | 64.27(+38.47) |
>   | OpenCLIP ViT-L-14 laion2b |  67.42 |        31.98 | 57.76(+25.79)  | 74.16(+42.19) |
>   | OpenCLIP ViT-H-14 laion2b |  68.87 |        32.19 | 58.44(+26.25)  | 74.64(+42.45) |
>   | UniCL_YFCC                |  30.73 |        16.26 | 30.99(+14.73)  | 42.06(+25.80) |
>   | UniCL_IN21K                  |  22.84 |        34.93 | 35.56(+0.63)   | 35.53(+0.60)  |
>   | UniCL_YFCC+IN21K             |  40.92 |        32.09 | 45.34(+13.25)  | 54.27(+22.18) |
>   | UniCL_all                 |  44.43 |        28.78 | 44.13(+15.35)  | 55.33(+26.55) |
>   | KLITE                     |  45.82 |        26.93 | 45.26(+18.33)  | 58.03(+31.10) |
>   | BLIP                      |  44.87 |        21.14 | 41.21(+20.07)  | 54.45(+33.31) |
>   | BLIP_ft                   |  45.88 |        23.62 | 43.99(+20.37)  | 57.12(+33.51) |
>   | FLAVA                     |  36.05 |        17.57 | 33.99(+16.42)  | 45.68(+28.10) |

---

### Meta-Review · Area_Chair_G3Kz · 2023-12-12

**Metareview:**

This paper was reviewed by five experts in the field. The paper received divergent ratings of 66338. The reviewers appreciated the importance of the problem that the paper is studying and the comprehensive experiments conducted. The major concerns gravitated in: 1) the findings in the paper are not surprising and straightforward (1Zb8, PhzX, pPRh, stdj, jHnf); 2) limited practical implications on how to improve VLMs and their downstream tasks (1Zb8, PhzX).

The AC read the paper, the reviews, and the authors' responses carefully. The AC agrees with the reviewers' assessments, especially the major concerns. The AC agrees with the authors that the community needs more papers focusing on evaluation and analysis, especially on widely used VLMs. However, the AC shares similar concerns that the findings in this paper provide limited knowledge advancement and insights to the community. The WordNet hierarchy was built based on linguistic granularity. The AC shares the concern with reviewer pPRh that improved performance on the proposed ImageNet-WordNet hierarchy benchmark may not translate to visual applications where VLMs are mostly used for. Furthermore, the ImageNet and COCO datasets are narrow in scope and small scale considering the current trend in the community of using VLMs as part of a foundation model. Finally, the paper doesn't provide potential directions for improving VLMs or their downstream tasks.

Therefore, the AC believes the current form of the paper doesn't meet the bar for a broader audience at ICLR. To make the paper stronger, the authors could strengthen the paper in one of the following ways: 1) evaluate on more comprehensive datasets to increase the practical implications of the conclusions. For example, the authors could conduct experiments on more vision-focused hierarchies such as LVIS and iNaturalist datasets; 2) provide some promising solutions to improve VLMs' performance on granularity and specificity. The authors could also consider resubmitting their work to a more focused track or workshop (e.g., NeurIPS Datasets and Benchmarks).

The authors are encouraged to consider the reviewers' comments when revising and resubmitting the paper.

**Justification For Why Not Higher Score:**

The AC shares the major concerns from the reviewers. The concerns around the limited insights and practical implications overweight the strengths of the paper.

**Justification For Why Not Lower Score:**

N/A

---

### Decision · Program_Chairs · 2024-01-16

Reject